# The Non-Essential Amino Acid Cysteine Becomes Essential for Tumor Proliferation and Survival

**DOI:** 10.3390/cancers11050678

**Published:** 2019-05-16

**Authors:** Joseph A. Combs, Gina M. DeNicola

**Affiliations:** Department of Cancer Physiology, H. Lee Moffitt Cancer Center, Tampa, FL 33612, USA

**Keywords:** cystine, cysteine, xCT, transsulfuration, glutathione, macropinocytosis, cystathionine, CBS, CSE, GGT

## Abstract

The non-essential amino acid cysteine is used within cells for multiple processes that rely on the chemistry of its thiol group. Under physiological conditions, many non-transformed tissues rely on glutathione, circulating cysteine, and the de novo cysteine synthesis (transsulfuration) pathway as sources of intracellular cysteine to support cellular processes. In contrast, many cancers require exogeneous cystine for proliferation and viability. Herein, we review how the cystine transporter, xCT, and exogenous cystine fuel cancer cell proliferation and the mechanisms that regulate xCT expression and activity. Further, we discuss the potential contribution of additional sources of cysteine to the cysteine pool and what is known about the essentiality of these processes in cancer cells. Finally, we discuss whether cyst(e)ine dependency and associated metabolic alterations represent therapeutically targetable metabolic vulnerabilities.

## 1. Introduction

It was recognized early in the 20th century that neoplastic tissues exhibit a dysregulated form of metabolism. Metabolic reprogramming is considered a hallmark of the oncogenic process [1], with much of cancer cell metabolism being dedicated to support high rates of proliferation and to alleviate associated cellular stress. It has long been hypothesized that this altered metabolism could be exploited to treat cancers [2]. The many metabolic changes in cancer cells may represent potentially targetable vulnerabilities [3].

Reliance on amino acids to fuel the high anabolic metabolism is a vulnerability of cancer cell metabolism. Amino acids are essential for oncogenic proliferation and survival [4,5,6,7]. Of particular interest are methionine and cysteine, the only proteinogenic amino acids among the many sulfur-containing amino acids, with the former being essential and the latter considered semi-essential. Both methionine and cysteine fulfill an important role as the main source of sulfur for a diverse set of biochemical reactions within the cell [8]. Recent studies suggest that the manner by which cancer cells obtain cysteine and how cysteine is used are key to cancer cell survival and more complex than previously appreciated. In this review, we discuss how cancers obtain cysteine, how this promotes cancer cell survival, and how this may ultimately represent an exploitable metabolic vulnerability.

## 2. Use of Cysteine

Cysteine is used widely throughout the cell for diverse roles including catalysis, trafficking, and mediating the oxidative stress response [9,10]. For a more in-depth overview of the uses of cysteine, please see Bak et al. and Stipanuk et al. [8,9]. The importance of cysteine within the cell lies within its sulfur moiety, which exists as a nucleophilic thiol (-SH) that is readily oxidized. When cysteine is incorporated into proteins, this reactivity helps to determine their form and function. Oxidation or electrophilic attack of the thiol by reactive molecules within the cell, such as reactive oxygen species (ROS), is used to maintain redox balance and in signaling. Cysteines within the protein also react to create disulfide bridges for proper protein folding. The reactivity of free thiols causes much of the available cysteine to be bound to other molecules but especially other cysteines by disulfide bonds. The oxidizing conditions of the extracellular environment promote cysteine oxidation, in which cysteine forms a dipeptide through disulfide bonding with another cysteine to form cystine. Cystine is the predominant form of cysteine extracellularly and a major form in which cysteine is acquired by tissues [11]. Alternatively, cysteine is produced from methionine through the de novo transsulfuration (TSS) pathway. Cysteine may also be salvaged from glutathione, taken up in its reduced form from extracellular sources, or from protein catabolism [12,13,14]. 

Cysteine usage is important in maintaining cellular homeostasis and survival in cancer cells. Although cysteine is incorporated into protein sparingly in a highly conserved manner due to its reactivity, protein synthesis accounts for the majority of cellular cysteine usage [9]. Another major use of cysteine within the cell is the production of the tripeptide antioxidant glutathione. Glutathione is composed of glycine, glutamate, and cysteine, with cysteine being limiting for glutathione synthesis in normal tissues. Glutathione is a key component in the cellular oxidative stress response through its direct oxidation of its thiol to produce oxidized glutathione (GSSG) and also through its use by enzymes such as glutathione peroxidases (GPx) [15]. Cancer cells contain high concentrations of glutathione for mitigating ROS generation and the detoxification of xenobiotics [16]. High glutathione levels buffer oxidative stress in cancer cells that would otherwise cause cell death. Cysteine is also used to produce the amino acid taurine which is used in mitochondrial function and control of cellular osmolarity, the gasotransmitter hydrogen sulfide (H_2_S), and iron-sulfur clusters for respiration and as a co-factor for various enzymes, including aconitase [17,18]. Use of cysteine to produce these compounds is dependent on cancer and microenvironmental context. For instance, taurine production is inhibited in some lung cancer cell lines allowing for a dramatic increase in intracellular cysteine [19]. Hydrogen sulfide production is increased in lung cancer cell lines to promote mitochondrial bioenergetics and for the control of mitochondrial DNA repair [20].

## 3. Regulation of Cysteine Metabolism

Cysteine is necessary to promote cancer cell proliferation and survival, and can be acquired through multiple pathways depending upon extracellular and intracellular conditions. The metabolic demands placed upon a cell from the stresses associated with proliferation caused by oncogenic transformation produce unique needs that must be met through extracellular sources of cysteine and de novo cysteine generation [5,21,22].

### 3.1. xCT Uptake of Cystine

Extracellular cystine is readily abundant (approximately 50 μM) within the body, as the liver produces and exports a great amount of cysteine that is quickly oxidized [11,23,24,25]. This glut of cystine is utilized by cancer cells to elevate intracellular cysteine levels for catabolic usage. Many cancer cell lines highly express the Na^+^-independent cystine/glutamate antiporter xCT, which is encoded by the *SLC7A11* gene [26,27,28,29] (Figure 1). The activity of this antiporter is attributed to system x_c_-. System x_c_- is composed of xCT and a separate protein, known as solute carrier family 3 member 2 (SLC3A2 or 4F2), that localizes xCT to the cell membrane [30]. For the purposes of this review, xCT will be used to refer to the activity and protein interaction ascribed to system x_c_-. 

The brain and immune system are the two major sites of xCT expression; however, xCT is dispensable for development [31]. However, the expression of xCT is highly inducible in normal cells and is likely a major component in dealing with stressors such as inflammation or infection that induce oxidative stress [27,31,32,33]. The import of cystine by xCT has been at the forefront of many cysteine metabolism studies in cancer cells as many cancers, including glioblastoma, triple-negative breast cancer, and non-small cell lung cancer, overexpress and use xCT for cystine uptake [22,26,27,28,29,34]. xCT expression drives increased uptake of cystine to produce cysteine for use by the cell [22,26,35]. 

The expression and activity of xCT are controlled by multiple factors, many of which are aberrantly active in cancer cells. Stress induced by starvation of key nutrients such as glucose or cysteine upregulates the transcription factors nuclear factor (erythroid-derived 2)-like 2 (NRF2) and activating transcription factor 4 (ATF4) to jointly or independently control xCT expression [36,37,38,39,40]. NRF2 is activated following exposure of cells to oxidative insult. Oxidization of cysteines on Kelch-like ECH-associated protein 1 (KEAP1) prevents NRF2 polyubiquitination and degradation [41]. NRF2 is subsequently stabilized to transcriptionally upregulate antioxidant response genes including xCT, the rate-limiting enzyme in glutathione synthesis, glutamate cysteine ligase (GCL), and enzymes used in detoxification of ROS, such as GPx [42], to attenuate ROS damage. Cystine uptake via xCT facilitates glutathione synthesis for ROS detoxification, thereby allowing restoration of KEAP1 function and redox homeostasis. Similar to NRF2, ATF4 is key in sensing and responding to cysteine availability. ATF4 is part of the integrated stress response (ISR) and responds to various stimuli such as endoplasmic reticulum (ER) stress or amino acid starvation. Under basal conditions, the interaction of eIF2 ternary complex and the start codon of the second upstream open reading frame (μORF2) in the 5’-untranslated region of ATF4 inhibits ATF4 translation [43,44]. Amino acid starvation promotes the accumulation of uncharged tRNAs that activate general control nonderepressible 2 (GCN2), which phosphorylates eukaryotic initiation factor 2α (eIF2α). Phosphorylated eIF2α represses global cap-dependent translation but allows cap-independent translation [44,45], resulting in translation of ATF4 and induction of ATF4 target genes. ATF4 and NRF2 target genes, including xCT, are essential in maintaining cellular homeostasis during amino acid deprivation by increasing mediators of amino acid uptake and stress response such as asparagine synthetase (ASNS), cationic amino acid transporter 1 (CAT1), NAD(P)H:quinone oxidoreductase (NQO1) and glutathione specific γ-glutamylcyclotransferase 1 (CHAC1) [38,46,47,48,49]. Amino acid starvation is not the only stressor capable of inducing xCT expression and activity. Stress induced through proteasomal inhibition, glucose starvation, glutamate toxicity, ER stress, or ROS generation activate NRF2 and ATF4 activity and upregulate xCT [36,37,39,50,51]. Transcription factors can also suppress xCT expression and oxidative stress responses. The tumor suppressor p53 negatively regulates xCT expression under stress conditions to promote ferroptotic cell death [52]. p53 is the most commonly mutated tumor suppressor, which may explain in part the frequent overexpression of xCT in human cancers. In addition, the tumor suppressor breast cancer 1 gene (BRCA1) associated protein-1 (BAP1), a deubiquitinase that is commonly silenced or lost in many cancers, also represses xCT expression [53]. Thus, there are multiple transcriptional inputs that control xCT expression that are commonly deregulated in cancer. 

xCT activity is also regulated through posttranslational modification. While some cancer cells are dependent on xCT for survival [15], a subset can limit xCT activity downstream of growth factor pathway signaling. Oncogenic phosphatidylinositol 3-kinase (PI3KCA) mutations activate protein kinase B (AKT), which phosphorylates xCT at serine 26 to suppress cystine/glutamate antiport [5]. Suppression of xCT activity induces a methionine dependency rather than a cystine dependency. This modulation of xCT activity by aberrantly active AKT may be specific to breast cancer cell lines and/or only present in cancer cells with oncogenic PI3KCA mutations as many cancer cells are cystine dependent. This site has also been shown to be a site of mechanistic target of rapamycin complex 2 (mTORC2) phosphorylation [22]. Oncogenic activation of AKT by mutant PI3KCA and mTORC2 signaling are common in cancers, and consequently xCT activity may be influenced by multiple posttranslational and metabolic inputs. Indeed, xCT is also regulated by substrate availability in the tumor microenvironment. Glutamate released by xCT into extracellular space accumulates and inhibits glutamate/cystine exchange. In triple negative breast cancer (TNBC) and non-small cell lung cancer (NSCLC) cells, inhibition of xCT by glutamate leads to a reduction in intracellular concentrations of free cysteine [34]. Free intracellular cysteine normally prevents auto-oxidation and inhibition of hypoxia-inducible factor prolyl hydroxylase 2 (EgIN1), which targets hypoxia inducible factor 1α (HIF1α) for degradation. Inhibition of xCT suppresses intracellular cysteine and stabilizes HIF1α [34]. In these cells, modulation of xCT is exploited to shape a tumor microenvironment conducive to increased aggressiveness. xCT activity is exploited to mediate both cell stress response and promote proliferation in many cancers. 

As a consequence of their dependency on xCT to acquire cysteine, cancer cells become vulnerable to modulation of xCT activity and metabolic processes related to xCT. As described previously, xCT activity requires obligatory export of glutamate for the import of extracellular cystine. Glutamate is produced intracellularly from glutamine via glutaminase, which supplies glutamate for both xCT activity and for tricarboxylic (TCA) cycle anaplerosis. High xCT activity in cells creates a glutamine/glutamate dependency that sensitizes cells to glutaminase inhibition [25,26,54]. This glutamine dependency relies on extracellular cystine content and xCT activity [25]. As noted previously, NRF2 regulates xCT expression. Tumors bearing a mutant form of KEAP1 unable to bind and degrade NRF2 are sensitive to glutaminase inhibition as a consequence of xCT activity [55]. KEAP1 mutant cells are glutamate deficient due to a high demand for glutamate for both glutathione synthesis and glutamate/cystine exchange, leading to inadequate glutamine entry into the TCA cycle [26]. Similarly, xCT expression induces glucose dependency via a similar TCA anaplerosis mechanism. Further, cystine reduction requires NADPH, which is primarily supplied via the pentose phosphate pathway (PPP) from glucose. With high rates of cystine uptake by xCT, NADPH quickly becomes limiting [19,37,54,56,57]. Thus, cystine-dependency rewires cellular metabolism to confer sensitivity to both glutamine and glucose starvation. 

Furthermore, xCT expression is an indication of extracellular cystine dependency in cancers. Inhibition, knockdown, or knockout of xCT remains a prime strategy to induce cancer cell death for therapeutic treatment [28,29]. Inhibitors of xCT have been used to exploit cancer cell reliance on exogenous cystine. Sulfasalazine (SAS) and erastin are two of the best characterized inhibitors of xCT activity [15,27,34,58,59]. Cell death induced by cystine starvation or blockade of xCT is mediated by the non-apoptotic cell death mechanism, ferroptosis [15,58,59,60]. For a comprehensive discussion of this topic, please see Stockwell et al. [59]. SAS is widely used because it is already in use in the clinic to treat rheumatoid arthritis [27]. Despite this, erastin is a more potent inducer of ferroptosis [15,58]. Ferroptosis by these small molecules is attributed to inhibition of both xCT activity and glutathione synthesis. Although glutathione depletion is essential for ferroptosis, inhibition of glutathione synthesis with buthionine sulfoximine (BSO) does not induce ferroptosis. The rate of glutathione depletion may play a role in this discrepancy, as active mitochondria deplete glutathione and lead to ferroptosis but inhibition of mitochondrial respiration spares glutathione and promotes cell survival [61]. A similar effect is seen by activation of wild-type p53. This downregulates xCT expression and activity but spares glutathione and leads to insensitivity to erastin treatment [62].

Although promising, xCT inhibition proved ineffective in a clinical trial for glioma therapy. SAS treatment was actively detrimental for patient survival and the trial was terminated early. Poor outcome possibly occurred because off-target effects of SAS exacerbated pre-existing conditions generated by the glioma [63]. An alternative to xCT inhibition is removal of its substrate, cystine. Cystine restriction would not be easy to achieve through dietary intervention in human patients. To address this challenge, Cramer et al. generated an engineered, stable form of cystathionine γ-lyase (CSE), called cyst(e)inase, capable of depleting cystine/cysteine from the circulation *in vivo*. Cyst(e)inase treatment effectively starved tumors, including NSCLC and prostate cancer, of extracellular cystine [6]. Unlike the observed toxicity of SAS in the clinic, cyst(e)inase treatment produced no toxic side effects following long term treatment in mice [6]. This study showed that many cancers are reliant on cyst(e)ine for survival. It argues that compensatory mechanisms of acquiring cysteine are not capable of sustaining survival in the tested cancers and indicates that xCT or other cyst(e)ine importers are relevant therapeutic targets. Given the importance of xCT in maintaining cancer cell survival, more studies are likely to follow targeting cystine and xCT. A current barrier for these studies is the lack of specificity of the drugs available to target xCT. Additionally, cystine starvation induces ferroptosis in many tested cancers; however, a subset of cancers is resistant. It remains to be seen whether use of compensatory pathways may explain how this resistance is achieved.

#### Other Cystine Transporters

While solute carrier family 3 member 1 (SLC3A1) is proposed to be a cysteine transporter, mutations in SLC3A1 lead to cystinuria [64,65], suggesting it instead transports cystine. SLC3A1 is regulated in a NRF2-dependent manner and is associated with cancer stem cells in liver cancer cell populations [66,67]. SLC3A1 upregulation also promotes breast cancer growth by the uptake of cyst(e)ine. The SLC3A1-mediated increase in intracellular cysteine ameliorates ROS and inhibits protein phosphatase 2A (PP2A) activity allowing AKT signaling to induce the activity of the transcription factor β-catenin [65]. Whether this same signaling cascade occurs through other cyst(e)ine importers or in other cell lines is unknown. SLC3A1 exit from the endoplasmic reticulum is facilitated by SLC7A9, and mutations in SLC7A9 also result in cystinuria [68]. Nothing is known of the importance of other cystine transporters in cancer cells.

### 3.2. The Transsulfuration Pathway and de Novo Cysteine Synthesis

An alternative source of cysteine to exogenous cystine is the TSS pathway. As detailed previously, cysteine produced by the TSS pathway is derived from methionine and serine. Methionine is obtained from the extracellular environment and converted into S-adenosylmethionine (SAM) by methionine adenosyltransferases (MAT) through the transfer of the adenosyl group from a molecule of adenosine triphosphate (ATP). SAM is the major source of methyl groups for methylation of biomolecules within the cell. These methylation reactions produce S-adenosylhomocysteine (SAH), which is subsequently metabolized to homocysteine (Hcy) by SAH hydrolase (SAHH). Hcy metabolism is the key branchpoint between the methionine cycle and the TSS pathway. Hcy may be remethylated to methionine by methionine synthase (MS), with 5-methyltetrahydrofolate donating the methyl group. Alternatively, Hcy methylation can be catalyzed by betaine homocysteine methyltransferase (BHMT), using betaine as the methyl donor. This reaction is tissue specific as BHMT is expressed primarily in the liver and kidneys and at low levels in the brain, testis, and lung [69]. Hcy may also irreversibly enter the TSS pathway through the condensation of serine and Hcy by cystathionine β-synthase (CBS) to produce cystathionine (Cth). Cth is subsequently hydrolyzed to cysteine, ammonia, and α-ketobutyrate by CSE [70] (Figure 1). 

Because methionine donates the sulfur, cysteine produced by the TSS pathway is derived from the methionine cycle. However, it is important to note that the methionine cycle contributes to other essential biological functions, including methylation of biological substrates. Because of this, only a fraction of cellular methionine is available for transsulfuration [71]. Regulation of methionine usage is tightly regulated via the levels of methionine cycle intermediates, which control the activity of the TSS pathway. CBS is a key point of regulation because of its position at the methionine cycle/TSS pathway branchpoint. Acting as an allosteric activator, SAM interacts with the C-terminal portion of CBS and promotes entry of Hcy into the TSS pathway, thereby irreversibly committing methionine sulfur to the synthesis of cysteine [8,70,71,72,73]. The SAM/SAH ratio approximately represents the methylation capacity of cells and CBS activity maintains this balance [8,74]. The C-terminal portion of CBS can also be cleaved to produce a 45 kDa constitutively and highly active form of CBS unresponsive to SAM regulation [8,39,75]. CBS cleavage has been observed following TNFα stimulation, and results in increased TSS pathway activity to promote glutathione synthesis to deal with ROS generation [76]. The presence of a SAM-independent form of CBS suggests that a metabolically uncoupled TSS pathway facilitates the rapid removal of Hcy or increased cysteine synthesis capacity, which promotes survival under oxidizing conditions. 

CBS and CSE are also regulated by multiple post-translational modifications and transcriptional control. Reviewed in depth by Sbodio et al., CBS activity is increased by glutathionylation, decreased by sumoylation, and altered by phosphorylation at serine 227 to produce more H_2_S than it would by canonical TSS pathway activity. CSE is activated by phosphorylation by AKT and expression is upregulated by farnesoid X receptor (FXR) and specificity protein 1 (Sp1) [39]. ATF4 and NRF2 regulate CBS and CSE transcriptionally, although CBS is upregulated and CSE is downregulated in ATF4-deficient mouse embryonic fibroblasts (MEF) suggesting that ATF4 control of the TSS pathway is dependent upon cellular and stimulus contexts [39,77]. Both CBS and CSE require the co-factor pyridoxal 5′-phosphate (PLP), the active form of vitamin B_6_, and CBS requires a heme group to function. CBS activity is induced following ER or oxidative stress conditions and exhibits a cytoprotective effect [10,78,79]. Functional CSE is crucial for ATF4-mediated cytoprotection by increasing the production of cysteine for glutathione synthesis [77].

Unlike xCT, almost all non-neoplastic tissues within humans and mice express CBS, CSE, or both of these proteins. The liver and pancreas express both CBS and CSE while the brain expresses CBS with little or no CSE expression or activity [80,81]. It is estimated that between 20–50% of the cysteine (available predominantly as cystine) within the body is produced in the liver through the TSS pathway [10,82]. Phenotypically, individuals harboring mutations in CBS all share elevated levels of methionine and homocysteine. Disease resulting from CBS mutations range from pneumothorax to mental retardation, thrombosis, ectopia lentis, and skeletal issues similar to Marfan syndrome [83,84]. This suggests that CBS activity is crucial in the cardiovascular tissues, brain, eye, skeletal muscle, and in bone growth. CBS complete knockout (KO) mice die post-natally within five weeks, while CSE KO mice are viable [85,86]. CSE KO mice do not have a notable phenotype in the presence of dietary cysteine. However, following cysteine starvation, these mice present with symptoms similar to CBS KO mice [86]. Like CBS mutations, CSE mutations result in elevated methionine and homocysteine as well as cystathionine [87]. The exact mechanism by which increased methionine, homocysteine, or cystathionine is of direct etiological significance in correlated disease states is controversial [88]. Treatment of humanized CBS mice, a model mimicking the reduced level of CBS activity caused by mutations in humans, with betaine reduces Hcy levels and ameliorates much of the toxicity induced by Hcy [79]. One potential mechanism of Hcy toxicity is a decrease in the SAM/SAH ratio that leads to hypomethylation and aberrant upregulation of genes within the cell [89,90]. The difference in severity of CBS and CSE KO mice phenotypes suggests that production of cysteine may not be the only important functions of CBS and CSE.

The inability to acquire cysteine through xCT may force cancer cells to use the TSS pathway, less common extracellular dipeptides, or extracellular glutathione to produce cysteine for proliferation [5,30,50,91]. A select group of cancer cells are capable of modulating xCT activity while relying on the TSS pathway for survival. Breast cancer cells harboring oncogenic PI3KCA mutations (E545K or H1047R) are dependent on extracellular methionine and cannot regenerate methionine from Hcy. These PI3KCA mutations activate AKT to phosphorylate xCT and suppress cystine/glutamate antiport activity [5]. The methionine dependency occurs at the branch point, Hcy, of the methionine cycle and the TSS pathway. The decreased cystine uptake leads to a dependency on the TSS pathway to produce intracellular cysteine. If methionine is removed and substituted with Hcy, Hcy is still committed to producing cysteine rather than being remethylated to produce methionine. As such, the cells become reliant on extracellular methionine for survival. As described previously, mTORC2 similarly regulates xCT activity, and suppresses cystine in glioblastoma, TNBC and NSCLC cell lines [22]. It is posited that mTORC2 regulation of xCT acts as a form of metabolic control. When cells are in replete or plentiful nutrient circumstances, xCT activity is suppressed by active mTORC2 allowing for anaplerotic glutamine usage and enhanced cellular growth. Inhibition of mTORC2 activity by nutrient deprivation causes export of glutamate by xCT to facilitate the uptake of extracellular cystine and the promotion of cellular survival in times of nutrient scarcity and cellular stress [22]. Additionally, by activating the TSS pathway and suppressing xCT, the susceptibility of cells to ferroptosis through xCT inhibition is decreased [5]. This may be accomplished by using other cellular mechanisms to acquire cysteine as well as slowing the depletion of glutathione. For example, wild-type p53 activation in fibrosarcoma cells slows the depletion of glutathione and inhibits xCT activity to prevent ferroptosis induction. This may be in response to stressors that preferentially induce ferroptosis rather than another cell death mechanism [62]. 

As aforementioned, many studied cancer cells are incapable of relying solely on the TSS pathway for survival. It is unknown why the TSS pathway is insufficient for the cysteine needs of these cells, but recent studies indicate that downstream use of the produced cysteine plays a role. Cysteinyl-tRNA synthetase (CARS) aminoacylates tRNAs with cysteine and its activity in fibrosarcoma cells is essential for xCT inhibition-induced ferroptosis. When CARS is knocked out, fibrosarcoma cells upregulate the TSS pathway and consequently become insensitive to xCT inhibition-induced ferroptosis. Ferroptosis sensitivity is reestablished by inhibiting CSE with propargylglycine (PPG) [91]. Other downstream processes that can rapidly use cysteine or cysteine products, like glutathione synthesis, cause stress in cysteine starved cells and mediate ferroptosis sensitivity [21,61,91]. Alternatively, many cancer cells may simply be unable to use the TSS pathway. For example, in hepatocellular carcinoma (HCC), downregulation of CBS is associated with disease severity and the CBS promoter is methylated in HCC, gastric, and colorectal carcinoma preventing expression [70,73,81,92]. This marks cancers like HCC as prime targets for xCT inhibition therapy as the TSS pathway may not be functional and able to compensate for loss of cystine import.

The TSS pathway and xCT both play integral roles in the ISR through ATF4 and the oxidative stress response through NRF2, suggesting that induction of cellular stress by traditional chemotherapeutics could open the possibility of targeting stressed cells through metabolic intervention [30,38,77]. It is unknown why some cancers are able to use the TSS pathway and are not reliant on xCT for cysteine. More work is needed to understand what distinct metabolic or signaling pathways separate the xCT-inhibition sensitive and insensitive cancers. Whether CBS and CSE have functional roles in cancer cell survival independent of their ability to produce cysteine from Hcy is unexplored. It remains to be seen whether the TSS pathway plays a role in tumorigenesis and cancer cell survival and whether it represents a viable therapeutic target. 

### 3.3. Cysteine-Specific Transporters

The majority of extracellular cysteine is oxidized to cystine, but cancers such as leukemia are capable of uptaking cysteine preferentially [12]. There is accumulating evidence for the presence of specific transporters for cysteine. Growth of ATF4 knockout MEFs requires the addition of the reducing agent β-mercaptoethanol, which reduces media cystine to cysteine, or excess cystine in the media to support their proliferation, viability and glutathione synthesis. Similarly, β-mercaptoethanol rescues cysteine starvation induced by xCT inhibition [52,91]. Additionally, extracellular breakdown of glutathione produces glutamate, glycine, and cysteine separately that are transported into the cell. These findings suggest that transporters capable of cysteine-specific uptake exist [13,93].

Little is known about cysteine-specific transporters, including their expression, mechanism of regulation, and whether they play important roles in cancer cell cysteine metabolism. A candidate cysteine-specific transporter is the excitatory amino acid transporter 3 (EAAT3), also referred to as SLC1A1 (Figure 2). EAAT3 is a member of a class of amino acid transporters notable for being expressed highly in the brain [94] and expressed in the lung, liver, kidneys, heart, and skeletal muscle [95]. EAAT3 can transport glutamate, aspartate, cysteine, and selenocysteine into the cell [94,96,97,98]. However, whether EAAT3 plays a role in cysteine metabolism in cancer cells is unknown.

Despite evidence for its existence, a transporter that can specifically take up cysteine and not cystine has not been positively identified in cancer cells. At approximately 10% of the concentration of cystine, the extracellular concentration of cysteine is low in comparison [99,100]. However, this is still a substantial amount of cysteine available for exploitation by cancer cells, and extracellular cysteine transport may play an unrecognized role in cancer cell survival. Given that a subset of cancers do not express xCT, more studies are necessary to determine whether a cysteine-specific transporter is necessary for the proliferation of these cancers.

### 3.4. Glutathione Degradation and Cysteine Salvage

Much like cystine, glutathione is produced in the liver and exported into the extracellular space [101]. The extracellular concentration of total glutathione ranges, depending on tissue and extracellular fluid analyzed, from approximately 3 μM to 400 μM in healthy adults [102,103,104]. Glutathione is critical in protecting tissues from oxidative stress and balancing the redox state of the extracellular environment but it can also serve as a source of cysteine for cells [13,105]. As the cysteine in glutathione is bound to glutamate and glycine as a tripeptide, glutathione must be degraded to make the cysteine freely available. The breakdown of glutathione to produce cysteine is achieved by the γ-glutamyl cycle. First, the extracellular facing γ-glutamyl transpeptidase (GGT) protein cleaves the γ-glutamyl bond of glutathione and liberates the γ-glutamyl amino acid from the cysteinylglycine dipeptide. The extracellular facing aminodipeptidase N then cleaves glycine from cysteine and allows cysteine to be taken up by cysteine transporters or oxidized and taken up by cystine transporters [13,93,106]. Alternatively, human peptide transporter 2 (PEPT2) can transport cysteinylglycine into the cell where it is cleaved by non-specific dipeptidases to free the cysteine [107] (Figure 2). 

Glutathione represents a small fraction of available extracellular cysteine but the γ-glutamyl cycle and GGT are essential for organismal development. GGT KO mice fail to thrive and die within 10–12 post-natal weeks. Their plasma contains 20% of the cysteine content of their wild-type counterparts. Supplementation with N-acetylcysteine reverses the phenotype of knockout suggesting that the lack of availability of cysteine from glutathione plays a major role in this phenotype [101]. In normal epithelial tissues, expression of GGT is localized to the ductal lumen where it breaks down glutathione for amino acid salvage [105]. However, the role of GGT in cancer cell cysteine metabolism is under-investigated. GGT is exploited in tumors as protein localization is lost, which allows for GGT to access glutathione in both the ductal space and in the interstitial fluid. Although cancers deriving from cells of ductal origin tend to express GGT, GGT expression closely correlates with the severity of the tumor regardless of tissue of origin [105]. Indeed, increased GGT expression is present in multiple types of cancers including nasopharyngeal, ovarian, and HCC. Further, overexpression of GGT in prostate cancer cells promotes proliferation in vivo [108,109,110]. The cysteine derived from glutathione is used intracellularly for the diverse roles previously discussed and as a means to reassemble glutathione intracellularly for oxidative regulation [13].

Similar to the extracellular use of glutathione to produce cysteine, cells are capable of degrading intracellular glutathione stores to produce cysteinylglycine through glutathione specific γ-glutamylcyclotransferase 1 (CHAC1) [38,111]. Expression of CHAC1 is mediated by the ISR. Amino acid starvation activates GCN2 subsequently inducing ATF4 expression which upregulates expression of CHAC1 [38,112,113]. As a counterbalance to CHAC1 activity, glutathione specific γ-glutamylcyclotransferase 2 (CHAC2) antagonizes the degradative activity of CHAC1 [114]. Depletion of intracellular glutathione by CHAC1 was initially considered pro-apoptotic but, in opposition to this, expression of CHAC1 in breast and ovarian cancers positively correlates with cancer grade and severity. CHAC1 overexpression in breast and ovarian cancer cells promotes proliferation and migration [38,111,115]. The shared ability of GGT and CHAC1 to degrade glutathione to cysteinylglycine suggests that CHAC1 may act as an intracellular cysteine scavenging mechanism when cells experience cysteine depletion. High concentrations of intracellular glutathione act as a cysteine reservoir. While possible that CHAC1 acts in a manner to scavenge cysteine from intracellular glutathione, knockdown of CHAC1 does not significantly alter cell death induced by erastin treatment [113]. Studies in GGT KO animals have demonstrated the importance of glutathione as a source of cysteine in the extracellular space [101]. It remains to be seen whether glutathione degradation and cysteine scavenging play a major metabolic role in cancers. These studies raise questions regarding the importance of glutathione as a cysteine source in cancers.

### 3.5. Macropinocytosis, Protein Scavenging, and Lysosomal Cyst(e)ine Transport

Proteins account for 70% of soluble substances in plasma with low molecular weight organic compounds, such as free amino acids, accounting for only 20% [116]. In standard in vitro tissue culture media, free amino acids are present in artificially high concentrations with added fetal bovine serum (FBS) providing only 5–10% of the free extracellular protein found in in vivo serum concentrations [117]. These conditions are reversed in vivo as extracellular protein concentration is roughly 200-fold greater than free amino acids [117]. Proteins in the extracellular space represent a large store of cysteine available for cells to use for proliferation and survival. The process of scavenging proteins from the extracellular space is accomplished through macropinocytosis. Macropinocytosis is the process by which a cell’s membrane protrudes to form cup-shaped ruffles and engulfs large amounts of extracellular fluid non-selectively [118,119]. Internalized macropinosomes containing proteins are shuttled to and fuse with lysosomes and the proteins are degraded into their constituent amino acids. These amino acids are then transported out of the lysosomes into the cytosol for use by the cell [119] (Figure 2). 

Macropinocytosis is promoted by onocogenic mutations in the RAS protein subfamily to promote the scavenging of protein as a supply of amino acids. This was first noted in bladder and pancreatic cancer cell lines both in vitro and in vivo [119]. Protein scavenging to resupply amino acids is also controlled by the mTOR and PI3K signaling pathways and transcriptional control of lysosomal/autophagic processes by the microphthalmia family (MiT/TFE) of transcription factors [120,121,122,123]. Under replete conditions, active mTORC1 and PI3K signaling through AKT provide a growth advantage to cancer cells by upregulation of free amino acid transporters and suppression of protein catabolism for amino acids [120,121]. However, even in replete conditions, macropinocytosed proteins provide a substantial amount of amino acids to the intracellular amino acid pools [117]. Protein catabolism for amino acids is aided by constitutive activity of MiT/TFE transcription factors in pancreatic cancer cells. Uncontrolled nuclear localization of MiT/TFE transcription factors increases the basal presence of lysosomes. Pancreatic cancer cells exploit constitutive MiT/TFE activity to buffer the stress of amino acid starvation through lysosomal protein catabolism [122]. 

Cysteine is found sparingly in proteins because of its reactivity, which would suggest that protein catabolism could be a relatively poor method to acquire cysteine [9]. However, intracellular cysteine levels are increased along with the other amino acids in pancreatic tumors in vivo from catabolism of labeled albumin [14]. Circumstantial evidence suggests that lysosomal degradation of protein is a possible source of intracellular cysteine. The disulfide bridges in proteins must be broken to permit peptidases to cleave proteins into amino acids. This is achieved by the import of intracellular cysteine into the lysosome via specific transporters [124]. Imported cysteine generates free cystine once the protein is broken down and must be exported from the lysosome and reduced in the cytosol to regenerate cysteine. This is achieved by the lysosomal cystine transport, cystinosin [125]. Mutations in the human cystinosin gene *CTNS* lead to cystinosis, a disease characterized by rickets, renal failure, growth retardation, and eventually death. Cystinosis is caused by an increased intracellular cystine concentration as cystine from protein catabolism in the lysosome is unable to exit and instead accumulates [126]. The mutations in cystinosin associated with the most severe form of cystinosis also poorly recruit mTOR to the lysosome [127]. Mutant cystinosin leading to deregulated mTOR signaling could play an unappreciated role in amino acid response. As cyst(e)inase should leave extracellular proteins untouched, the effectiveness of cyst(e)inase treatment against several cancers in vivo suggests that either macropinocytosis is not active in these cancers or that cysteine is not derived from protein catabolism at high enough rates to support cell survival [6]. Whether protein catabolism through macropinocytosis is a viable source of intracellular cysteine is currently unknown. Future studies may shed light on the importance of cysteine from macropinocytosis and protein catabolism as cystinosin is integral in the survival and function of normal tissue [125,126]. 

## 4. Conclusions and Remaining Questions

Cysteine metabolism in cancer cells is far more complex than once appreciated. The ability of cancers to modulate the manner in which they acquire cysteine to promote proliferation may involve key selective events during oncogenesis. Why some cancers are able to rely on both xCT mediated cysteine accumulation and the TSS pathway has been only somewhat explored [128]. Whether the TSS pathway, GGT, cysteine transporters, or macropinocytosis play important roles in tumorigenesis and cell survival in cancers with xCT activity is an open question. The TSS pathway has received relatively minor attention in the study of cancer biology as compared to xCT. The importance of the TSS pathway even in normal extrahepatic tissues is not well understood and studied [6,39,70,128,129]. GGT and cysteine-specific transporters have received even less attention. 

An important aspect of future in vitro studies is determining the proper in vivo nutrient levels of the tumor microenvironment and growing cells in media mimicking this [130,131]. Current in vitro tissue culture media contains extremely high levels of almost exclusively cystine. The concentrations of thiols in vivo are likely to contain a variable mixture of oxidized and reduced glutathione, cysteine, cystine, and proteins. Culture of cancer cells in media mimicking human plasma substantially alters cancer cell metabolism, and, to complicate the matter, nutrient levels in tumor interstitial fluid are different than the nutrient levels found in plasma due to poor vascular architecture and heterogeneity in the tumors [130,131]. Cystine dependency has been extensively studied using non-physiological tissue culture media, so whether media more closely matching conditions experienced by tumors in vivo would provide greater insight to cystine dependency in cancer cells is an unknown but intriguing question.

Nevertheless, xCT expression is quite common in cancers and the results of the cyst(e)inase study by Cramer et al. argue that in many cancers extracellular cyst(e)ine is necessary and the TSS pathway is inadequate for survival [6]. Failure of clinical trials to show benefit in targeting xCT in gliomas with SAS argue that better drugs are required before this metabolic vulnerability is exploited therapeutically [63]. Another limitation in our understanding of xCT necessity in cancers is a lack of a conditional mouse *SLC7A11* knockout allele to study tumor specific deletion of xCT in vivo. This model would enable studies to determine whether compensatory methods, such as glutathione degradation or the TSS pathway, play any role in cancer survival. It could also help to determine whether a potent and specific pharmacological inhibitor of xCT is a practical therapeutic option against this vulnerability.

Given that different cancers experience different stressors, and that even within tumors different cell populations experience varied stressors, it is likely that control of where and how these cells obtain cysteine is controlled for survival and proliferation. This is further complicated when considering interaction of the tumor with the tumor microenvironment. For example, Burkitt lymphoma and lymphocytic leukemia derive their cysteine almost exclusively from their surrounding stromal tissue [12,132]. Tumor microenvironment and cancer cell crosstalk may determine usage and generation of cysteine in an as yet undiscovered manner and further highlight the need for genetically engineered mouse models (GEMM). Metabolomics studies have the ability to combine new techniques to stabilize cysteine and interrogate metabolic states within cancer cells. Cysteine is highly reactive, making in vivo determination of cysteine metabolism difficult but new and adapted techniques are available to explore whether in vitro observations hold true in vivo [129]. With a growing understanding of the importance of cysteine in cellular processes, future studies will provide insights into whether cysteine metabolism represents a therapeutically targetable cancer metabolic vulnerability.

## Figures and Tables

**Figure 1 cancers-11-00678-f001:**
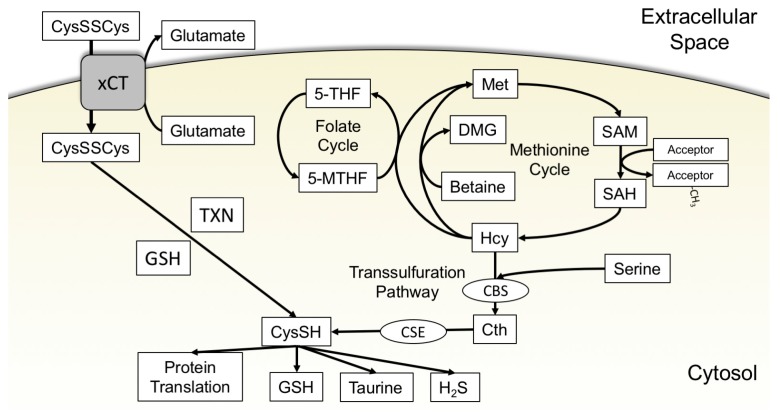
Acquisition of cysteine from extracellular cystine through xCT and de novo production of cysteine through the transsulfuration pathway. (Left) Extracellular cystine is transported into the cell through xCT while glutamate is exported. Thioredoxin or glutathione reduce cystine to cysteine, which is subsequently used for the synthesis of proteins, glutathione, and other sulfur-containing molecules. (Right) The de novo cysteine synthesis pathway (transsulfuration) requires the sulfur group from methionine. In the methionine cycle, methionine is adenosylated to produce SAM. SAM donates a methyl group to a methyl acceptor (protein, RNA, or DNA) to generate SAH. SAH hydrolase (not shown) generates Hcy from SAH. Hcy can regenerate methionine by accepting a methyl group from betaine or 5-MTHF from the folate cycle. Hcy can alternatively exit the methionine cycle via its condensation with serine by CBS in the first step of the transsulfuration pathway to produce Cth. Cth is subsequently hydrolyzed by CSE to ammonia (not shown), α-ketobutyrate (not shown), and cysteine. 5-MTHF: 5-tetramethylhydrofolate, 5-THF: 5-tetrahydrofolate, Cth: cystathionine, CBS: cystathionine β-synthase, CSE: cystathionine γ-lyase, CysSH: cysteine, CysSSCys: cystine, DMG: dimethylglycine, GSH: glutathione, Hcy: homocysteine, Met: methionine, SAH: S-adenosylhomocysteine, SAM: S-adenosylmethionine, TXN: thioredoxin.

**Figure 2 cancers-11-00678-f002:**
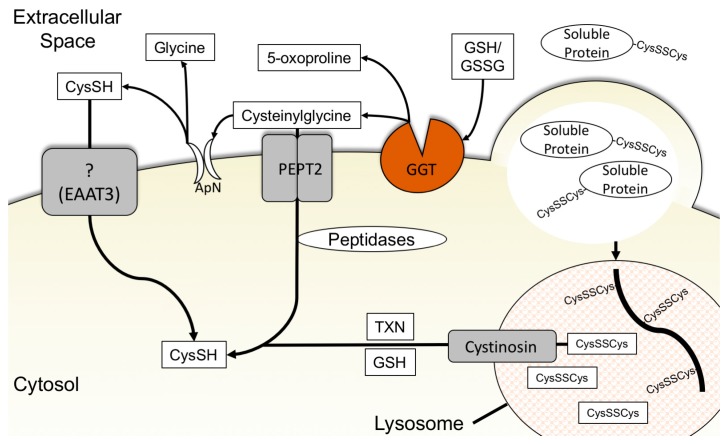
Cells acquire cysteine from extracellular glutathione through the γ-glutamyl cycle, cysteine uptake, and lysosomal protein scavenging by macropinocytosis with cystine export. (Left) Extracellular cysteine is transported into the cell through unidentified cysteine transporters, possibly EAAT3. (Center) The γ-glutamyl bond in extracellular glutathione is cleaved by GGT to create 5-oxoproline and the dipeptide cysteinylglycine. Cysteinylglycine can be taken up by the cell through PEPT2 and cleaved intracellularly by non-specific dipeptidases to cysteine and glycine. Cysteinylglycine may also be cleaved to cysteine and glycine extracellularly by ApN, followed by the cellular uptake of cysteine by cysteine transporters or cystine via xCT (not shown). (Right) Soluble proteins in the extracellular fluid are engulfed by the cell through macropinocytosis. The macropinosome containing proteins travels to and fuses with the lysosome. The disulfide bridges of the proteins are broken, and the protein is linearized by interaction with cysteine imported into the lysosome. The protein is hydrolyzed into its constituent amino acids, including cystine. Lysosomal cystine is subsequently exported to the cytosol by the lysosomal cystine transporter, cystinosin, and reduced by thioredoxin or glutathione to cysteine. ApN: Aminopeptidase N, CysSH: cysteine, CysSSCys: cystine, EAAT3: excitatory amino acid transporter 3, GGT: γ-glutamyltransferase, GSH: glutathione, GSSG: oxidized glutathione, PEPT2: peptide transporter 2, TXN: thioredoxin.

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
