# Peer review of "The Non-Essential Amino Acid Cysteine Becomes Essential for Tumor Proliferation and Survival"

_cancers, 2019, doi:10.3390/cancers11050678_

Round 1
Reviewer 1 Report
The review by Combs and DeNicola focuses on the role of cysteine in cancer metabolism and tumor biology. This is an important area of research in the field and is an extremely critical topic that has not been extensively reviewed until now. The review is very well written and covers many topics thoroughly and in an unbiased fashion. There are only very minor suggestions, as listed below:
Line 41. Would be good to change "sulfur molecule" to "sulfur moiety" or "sulfur atom"
Line 94, "excit" to "exit"
Line 95, "cyscle" to "cycle"
Line 357. Would be good to add comma to "broken,"
Line 420. Addition of FBS to media adds protein, although approximately to 5-10% of what is observed in vivo. May want to consider revising this sentence slightly to reflect this.
Author Response
Response to Reviewer 1 Comments
We appreciate the feedback from Reviewer 1 and have made the following changes and amendments.
Response 1: “Sulfur molecule” on Line 41 has been changed to “sulfur moiety” to be more technically correct in describing the chemistry highlighted in this sentence.
Response 2: “Excit” has been corrected to “exit” on Line 94
Response 3: “Cyscle” has been corrected to “cycle” on Line 95
Response 4: A comma has been added to “broken” on Line 361
Response 5: Sentences starting on Line 424 have been revised to better clarify that standard in vitro tissue culture media is not devoid of protein. This has been rectified by noting that the addition of FBS in in vitro tissue culture media adds a small amount of protein.

Reviewer 2 Report
This is a comprehensive review on cancer Cysteine metabolism by well-qualified authors. It is well-written and neatly organized. It described the regulation of Cysteine metabolism through Cystine transportation, de novo synthesis and salvage in great detail and highlighted the potential of targeting each component for cancer therapy. The figures are clear and informative as well. Overall, the review is easy to follow and will provides readers with important and updated knowledge on Cysteine metabolism. I only have minor suggestions.
1. xCT expression is upregulated in many cancers. On the other hand, growth factor signaling-mediated AKT activation is also common in most cancer types. Since AKT activation may suppress xCT activity, it would be better to add one or two sentences to clarify whether AKT-dependent xCT phosphorylation/inhibition is relevant in most cancer types or only for those with AKT hyperactivation due to oncogenic PI3K mutations.
2. Line 62, did the authors mean ‘glutathione levels buffer oxidative stress’?
3. Line 113, should it be ‘NRF2 is a key component in maintaining intracellular cysteine’?
4. Line 502, the authors should spell out GEMM.
Author Response
Response to Reviewer 2 Comments
We appreciate the feedback from Reviewer 2 and have made the following changes and amendments.
Response 1: As xCT activity and cystine dependency is the focus of the portion of the review within Lines 144-152, sentences have been added to better clarify the role of AKT activity in relation to xCT activity. Oncogenic PI3KCA mutation driving AKT to modify xCT activity in a subset of cancer cells has been emphasized to clarify that AKT modulation of xCT activity is not seen in all cancer cells.
Response 2: Line 62 has been corrected to state that glutathione buffers oxidative stress and not generic “stress.”
Point 3: Lines 113-119 have been revised for clarity. A more accurate description of how ROS interacts with KEAP1 and activates NRF2 activity has been included. This description includes a clearer explanation of how NRF2 controls cysteine levels in response to oxidative insults.
Response 4: GEMM has been spelled out on Line 508
